# Contributions of Non-Neuronal Cholinergic Systems to the Regulation of Immune Cell Function, Highlighting the Role of α7 Nicotinic Acetylcholine Receptors

**DOI:** 10.3390/ijms25084564

**Published:** 2024-04-22

**Authors:** Koichiro Kawashima, Masato Mashimo, Atsuo Nomura, Takeshi Fujii

**Affiliations:** 1Department of Molecular Pharmacology, Kitasato University School of Pharmaceutical Sciences, Minato-ku, Tokyo 108-8641, Japan; 2Department of Pharmacology, Faculty of Pharmaceutical Sciences, Doshisha Women’s College of Liberal Arts, Kyotanabe 610-0395, Japan; mmashimo@dwc.doshisha.ac.jp (M.M.); a-nomura@dwc.doshisha.ac.jp (A.N.)

**Keywords:** local regulator, non-neuronal, cholinergic system, immune cell, α7 nAChR, T cell, differentiation, Treg

## Abstract

Loewi’s discovery of acetylcholine (ACh) release from the frog vagus nerve and the discovery by Dale and Dudley of ACh in ox spleen led to the demonstration of chemical transmission of nerve impulses. ACh is now well-known to function as a neurotransmitter. However, advances in the techniques for ACh detection have led to its discovery in many lifeforms lacking a nervous system, including eubacteria, archaea, fungi, and plants. Notably, mRNAs encoding choline acetyltransferase and muscarinic and nicotinic ACh receptors (nAChRs) have been found in uninnervated mammalian cells, including immune cells, keratinocytes, vascular endothelial cells, cardiac myocytes, respiratory, and digestive epithelial cells. It thus appears that non-neuronal cholinergic systems are expressed in a variety of mammalian cells, and that ACh should now be recognized not only as a neurotransmitter, but also as a local regulator of non-neuronal cholinergic systems. Here, we discuss the role of non-neuronal cholinergic systems, with a focus on immune cells. A current focus of much research on non-neuronal cholinergic systems in immune cells is α7 nAChRs, as these receptors expressed on macrophages and T cells are involved in regulating inflammatory and immune responses. This makes α7 nAChRs an attractive potential therapeutic target.

## 1. Introduction

In 1914, Ewins [1] identified acetylcholine (ACh) as an active principle in ergot, a product of the fungus *Claviceps purpurea*, that elicits an inhibitory effect on the heart but a stimulatory effect on intestinal muscle. This discovery led Dale to study the pharmacological properties of various choline esters, including ACh, and found that ACh produced effects most similar to muscarine [2]. However, because of the lack of evidence for the presence of ACh in the animals, Dale was reluctant to suggest that ACh may be a neurotransmitter in parasympathetic neurons (see a review by Burgen [3]). In 1921, Loewi found that electrical stimulation of the attached vagus nerve induced release of Vagusstoff, a factor exerting a negative chronotropic effect, from isolated frog hearts into the perfusate [4]. Subsequently, Vagusstoff was pharmacologically identified as ACh [5]. In 1929, Dale and Dudley discovered ACh in the spleens of oxen and horses [6]. Based on those findings, Dale proposed the term “cholinergic” to describe nerves that transmit their action through release of ACh [7]. The Nobel Prize was awarded to Sir Henry Dale and Otto Loewi in 1936 for their groundbreaking discovery of chemical transmission of the effects of nerve impulses. At present, ACh is a well known classic neurotransmitter in the central and peripheral cholinergic systems. It is important to note, however, that although it was not known at the time, the spleen is not innervated by cholinergic nerves [8].

Here, we will first discuss the major components of cholinergic systems: (1) ACh; (2) choline acetyltransferase (ChAT, E.C. 2.3.1.6), an ACh-synthesizing enzyme; and (3) muscarinic and nicotinic ACh receptors (mAChRs and nAChRs, respectively).

### 1.1. ACh

ACh is relatively stable when kept in acidic solutions with a pH between 3.5 to 5.5 and is chemically unstable in neutral or alkaline solutions, even when stored at −20 °C in a freezer. Furthermore, ACh is highly unstable within a physiological context. For example, the effects of ACh released from nerve terminals at the neuromuscular junction are terminated within a few milliseconds by the action of the ACh-degrading enzyme acetylcholinesterase (AChE) [9]. Consequently, levels of ACh in tissues and biological fluids other than the brain are very low and were difficult to detect until the development of a sensitive and specific radioimmunoassay (RIA) for ACh [10,11] and high performance liquid chromatography-electrochemical detection (HPLC-ECD) [12].

### 1.2. ChAT

Within cholinergic neurons, ACh is synthesized in the cytosol from choline and acetyl-CoA through a reaction catalyzed by ChAT [13]. When measuring ChAT activity in peripheral tissue homogenates using the Fonnum procedure [14], it is important to keep in mind that both ChAT and the mitochondrial matrix enzyme carnitine acetyltransferase (CarAT, E.C.2.3.1.7) can catalyze ACh synthesis [13,15]. Thus, ACh synthesizing activity in peripheral tissue homogenates determined using the Fonnum procedure represents the sum of these two enzyme activities [14]. To determine specific ChAT activity, it is necessary to assess ACh synthesis in the presence of bromo-ACh (BrACh) or bromo-acetylcarnitine (BrACar), specific inhibitors of ChAT and CarAT, respectively [13,15]. The physiological significance of ACh synthesized by CarAT is still unknown.

Recent findings suggest the presence of significant ACh-synthesizing activity in plasma [16,17]. However, that activity was determined in the presence of abundant acetyl-CoA [16,17], which is unstable and thus virtually unavailable in plasma under normal extracellular conditions [18,19]. Therefore, the physiological significance of plasma ACh-synthesizing activity is currently unclear.

### 1.3. mAChRs and nAChRs

Various subtypes of mAChRs and nAChRs are expressed in non-neuronal cholinergic tissues and organs.

#### 1.3.1. mAChRs

Five distinct mAChR subtypes (M_1_–M_5_), acting via two different second messenger signaling systems, have been identified through molecular cloning [20,21]. M_1_, M_3_, and M_5_ mAChRs are coupled to G_q/11_, which, upon stimulation, mediates activation of phospholipase C (PLC), leading to increases in the intracellular free Ca^2+^ concentration ([Ca^2+^]_i_). M_2_ and M_4_ mAChRs are coupled to G_i/o_, which, upon stimulation, mediates inhibition of adenylyl cyclase, leading to declines in cAMP production [22]. Most non-neuronal cholinergic tissues and cells express all five mAChR subtypes [23,24,25].

#### 1.3.2. nAChRs

Non-neuronal cholinergic cells and tissues express primarily neuron-type nAChRs composed of only α and β subunits: eight α (α2–α7, α9–α10) and three β (β2–β4) [23,24,25]. α7 nAChRs, composed of homomeric α7 subunits, are currently the subject of much research on non-neuronal cholinergic systems (see Section 4). In addition, subunit α9 assembles with α10 to form heteromeric α9α10 nAChRs, which exhibit high Ca^2+^ permeability in auditory hair cells [26] and human and murine monocytes [27].

## 2. Expression of ACh in a Wide Variety of Life Forms

As mentioned above, Ewins in 1914 [1] identified ACh as a biologically active principle in ergot, and showed that the fungus has the ability to produce ACh. Since then, sporadic papers have been published reporting the detection of ACh in eubacteria, unicellular animals, and other organisms (see reviews [28,29,30]). Taking advantage of the sensitivity, specificity, and operational simplicity of an RIA for ACh, Horiuchi et al. [31,32] and Yamada et al. [31,32] conducted comparative biological studies examining the ACh content and ACh-synthesizing activity in various life forms (Figure 1) [33].

They found that all the life forms tested, which included various eubacteria, archaea, fungi, plants, and animals, expressed some level of ACh and ACh-synthesizing activity. At present, the enzymes catalyzing ACh synthesis and governing expression of AChRs in eubacteria, archaea, fungi, and plants remain unknown [34]. However, “17 Shiitake mushroom” and “18 Bamboo shoot in both upper and lower portion”, as well as “16 Insects and Annelids”, exhibit ChAT-like ACh-synthesizing activity that is sensitive to BrACh, a selective ChAT inhibitor. Additionally, “9 Thermoccales (*T. kodakaraensis* KOD1) collected during the rapid growth phase showed detectable levels of ChAT-like ACh-synthesizing activity.

These and other findings suggest that ACh is an evolutionarily ancient molecule that functions as a mediator between adjacent cells and plays a role in regulating growth, differentiation, water homeostasis, or photosynthesis in archaea [32], mushrooms and rapidly growing bamboo shoots [31], maize sprouts [35,36], and Urtica dioica [28,29]. However, understanding the biological function of ACh in life forms other than animals will require further research. This is important, as expression of significant levels of ChAT-like activity in rapidly growing bamboo shoots suggests that creation of transgenic plants that overproduce ACh may promote faster growth and higher yields of crops that could potentially help reduce atmospheric carbon dioxide and solve food shortages.

## 3. Expression of Non-Neuronal Cholinergic Systems in Mammalian Species

In 1978 Sastry and Sadavongvivad [37] published a pioneering review on cholinergic systems in non-neuronal tissues. Since then, research in molecular biology has accelerated the development of tools for determining ACh (e.g., HPLC-ECD and RIA) and for detecting mRNA expression of cholinergic components. Correspondingly, evidence of expression of non-neuronal cholinergic systems in various mammalian tissues and cells has been accumulating. Several reviews addressing this topic in general, and in specific areas are currently available (see reviews [23,24,25,38,39,40,41]. As a result, “non-neuronal ACh or cholinergic systems” are now widely recognized in the field of biology. Representative examples of non-neuronal mammalian cells and tissues expressing ACh or ChAT are listed in Table 1.

## 4. Cholinergic System in Immune Cells

It has long been known that T cells, B cells, macrophages and dendritic cells all express both mAChRs and nAChRs and that mAChR and nAChR agonists elicit various biochemical and functional effects (see reviews [59,60,61]). Based on those observations, it was thought until the early 1990s that the parasympathetic nervous system might be involved in neuro-immune crosstalk [62,63,64]. However, using the aforementioned advanced techniques, we were able to show that immune cells express all the components necessary to compose a cholinergic system, including ACh, ChAT, mAChRs and nAChRs, and AChE (see reviews [59,60,61,65,66]).

### 4.1. ACh Synthesis in Immune Cells

The search for the origin of ACh detected in human plasma [67] led to studies measuring the ACh concentrations in blood cell fractions and plasma [68]. Those studies revealed that the ACh concentration in rabbit blood cells was 25 times higher than in plasma. Moreover, intravenously injected nicotine elicited a significant increase in plasma ACh and a decrease in the ACh content of blood cells, suggesting release of ACh from blood cells to the plasma, while expression of ChAT activity was detected in rabbit immune cells and rat lymphocytes [69]. In addition, Kawashima et al. [70] found that the stable amounts of ACh in human peripheral blood were localized in mononuclear leukocytes (MNLs) consisting mainly of lymphocytes and a small fraction of monocytes (Figure 2A(1)). Although considerable interindividual variation in the ACh content of whole blood was observed, little variation was seen in a given individual when determined on two different occasions between 6 and 24 months apart (Figure 2A(2)). Furthermore, the ACh content in MNLs (about 60% of the whole blood) correlated well with that in whole blood (Figure 2B). These results indicate that plasma ACh is derived from MNLs, which suggests that immune cells have the ability to synthesize and release ACh. The relationship between the observed interindividual variation in immune cell ACh content and immunity has not yet been elucidated. Considering the stable ACh content of MNLs within individuals, it would be interesting to investigate the relationship between the ACh content of immune cells and gut microbiota.

#### 4.1.1. ChAT in Immune Cells

##### Determination of ACh Synthesizing Activity in Immune Cells

Fujii et al. [48] detected varying contents of ACh in human leukemic cell lines used as models for human T cells, B cells, and monocytes. In addition, using BrACh and BrACar, specific inhibitors of ChAT and CarAT, respectively, they found that ACh synthesis catalyzed by CarAT was higher than that catalyzed by ChAT in these cell lines. However, phytohemagglutinin (PHA), which activates MOLT-3 human leukemic T cells via T-cell receptor (TCR)-mediated pathways, enhanced ChAT activity but not CarAT activity [48]. These observations suggest that ChAT expression in T cells is regulated by immune activity, whereas CarAT is not involved in regulating immune function. Therefore, before attributing ACh-synthesizing activity measured in immune cells with the Fonnum method to ChAT, it is recommended to confirm the BrACh sensitivity of that activity.

#### 4.1.2. ChAT mRNA and Enzyme Expression in Immune Cells

Using reverse transcriptase-polymerase chain reaction (RT-PCR) and Western blot analysis for ChAT protein, Fujii et al. [71] were the first to detect ChAT mRNA and protein expression in the MOLT-3 T cell line, and were the first to directly demonstrate the involvement of ChAT in ACh synthesis in immune cells. Later, expression of ChAT mRNA was similarly detected in human circulating lymphocytes [48,72,73], rat T and B cells [49] and MNLs [74], and mouse dendritic cells [75]. Expression of ChAT mRNA and protein in mature and immature human dendritic cells was subsequently confirmed using RT-PCR and immunocytochemical analyses [76].

ChAT expression in mouse T and B cells, dendritic cells, and macrophages was later further confirmed using ChAT^BAC^-eGFP transgenic mice [77] and ChAT-Cre-tdTomato mice [78]. These findings indicate that immune cells express ChAT for ACh synthesis.

#### 4.1.3. Mechanisms Regulating ACh Synthesis and Release

As described above, T cell stimulation with PHA via a TCR-mediated pathway enhances ACh synthesis by promoting ChAT mRNA expression and increasing ChAT activity [72]. Phorbol 12-myristate 13-acetate (PMA), a protein kinase C (PKC) activator, and dibutyryl cAMP (dbcAMP), a protein kinase A (PKA) activator, also increase ChAT activity and ACh synthesis in MOLT-3 human leukemic T cells by upregulating ChAT gene expression [79,80]. Similarly, the calcium ionophores A23187 and ionomycin also upregulate expression of ChAT mRNA and its activity [81]. By contrast, FK506, an immunosuppressant calcineurin inhibitor, suppresses PHA-induced upregulation of ChAT mRNA expression [80]. These findings suggest the involvement of calcineurin-mediated pathways in ChAT gene transcription. Moreover, the summarized data provide compelling evidence that T cell activation during immune responses mediated through PKC-MAPK and/or adenylate cyclase-cAMP pathways upregulate ACh synthesis and release and suggest the lymphocytic cholinergic system is involved in regulating immune function [79,80].

The α7 nAChR allosteric ligand SLURP-1 promotes ChAT mRNA expression in MOLT-3 human leukemic T cells. This effect is abolished by the α7 nAChR antagonist methyllycaconitine, suggesting a role for α7 nAChRs in regulating ChAT expression in T cells [82].

Recent findings also indicate that during lymphocytic choriomeningitis virus infection, ChAT gene expression is markedly induced in CD4 and CD8 T cells in an IL-21-dependent manner, which promotes T cell migration [83].

#### 4.1.4. Storage and Release of ACh in Immune Cells

Within cholinergic neurons, ACh is synthesized in the cytosol and then transported by vesicular ACh transporter (VAChT) into synaptic vesicles, where it is stored at about 100 times the cytosolic concentration [84]. The VAChT gene is located within the first intron of the ChAT-encoding gene and is in the same transcriptional orientation as ChAT [85,86,87], and coordinated upregulation of VAChT and ChAT gene expression is observed in cholinergic neurons [84,88,89,90,91]. However, no VAChT mRNA expression has been detected in human peripheral blood MNLs, including T cells, B cells, and monocytes, even after stimulation with PHA to promote ChAT mRNA expression [72]. These findings argue against vesicular storage of ACh in lymphocytes.

Mediatophore is a homo-oligomer of a 16-kDa subunit homologous to proteolipid subunit c of vacuolar H^+^-ATPase. Mediatophore has been shown to translocate ACh [92,93,94] and to be involved in quantal ACh release at the *Torpedo* nerve-electroplaque junction [95]. Expression of mediatophore mRNA has been confirmed in CCRF-CEM and MOLT-3 human leukemic T cells [96]. PHA upregulates both ChAT and mediatophore mRNA expression, thereby increasing ACh release [96]. Furthermore, anti-mediatophore siRNA downregulates expression of mediatophore mRNA and decreases ACh release, but does not suppress expression of ChAT mRNA [96]. These findings suggest that T cells express mediatophore, which then plays an important role in mediating ACh release, and that mediatophore expression is regulated via TCR-mediated pathways.

### 4.2. Expression of AChRs in Immune Cells

As early as the 1970s, evidence of expression of mAChRs and nAChRs was detected in lymphocytes isolated from mouse, rat, and human thymus, lymph node, spleen, and peripheral blood in studies examining the induction of functional and biochemical effects of cholinergic ligands [63] (see also reviews [59,60,61]). Later, the expression of various mAChR subtypes and nAChR subunits in human and mouse lymphocytes was confirmed using RT-PCR [75,97] (see also a review [98]) (Table 2A). In C57BL/6J mice, mRNAs encoding all five mAChR subtypes are expressed in MNLs, dendritic cells, and macrophages [75,97]. In humans, however, the pattern of mRNA expression of each mAChR subtype varies somewhat among individuals, probably due to differences in their immunological status.

#### 4.2.1. mAChRs

All five M_1_–M_5_ mAChR subtypes are expressed to varying degrees in human and mouse immune cells [75,97,98]. The mAChR agonist oxotremorine (Oxo)-M causes an increase in [Ca^2+^]_i_ and subsequent [Ca^2+^]_i_ oscillations in human leukemic T and B cell lines (CEM and Daudi cells, respectively), leading to potentiation of c-fos gene expression [99,100]. This finding suggests ACh released from T and B cells acts on M_1_, M_3_, and/or M_5_ mAChRs to induce intracellular Ca^2+^ signaling that triggers nuclear signaling and upregulates gene expression.

M_1_/M_5_ mAChR gene-deficient (M_1_/M_5_-KO) mice exhibit lower concentrations of plasma anti-ovalbumin (OVA)-specific IgG_1_ antibodies when immunized with OVA than wild-type (WT) mice [101]. Furthermore, upon stimulation with OVA, spleen cells from M_1_/M_5_-KO mice immunized with OVA secrete smaller amounts of the proinflammatory cytokines tumor necrosis factor (TNF)-α, interferon (IFN)-γ, and interleukin (IL)-6 than spleen cells from WT mice [101]. These findings suggest that M_1_ and/or M_5_ mAChRs are involved in regulating pro-inflammatory cytokine production leading to the modulation of antibody production.

In human lung macrophages expressing M_1_ and M_3_ mAChR mRNAs, the non-specific AChR agonist carbachol promotes release of the proinflammatory mediator leukotriene B4, and that effect is attenuated by the selective M_3_ mAChR antagonist 4-DAMP [102]. These findings suggest that M_3_ mAChRs on macrophages are involved in regulating immune function by promoting proinflammatory mediator and cytokine production.

#### 4.2.2. nAChRs

Immune cells express nAChRs composed mainly of α2–α7, α9, α10, β2, and β3 neuronal type subunits [75]. In human T cells and B cells, for example, variable expression patterns have been observed for mRNAs encoding the α3, α5, α7, α9, and α10 subunits (see Table 2B [98]). By contrast, in C57BL/6J mice, mRNAs encoding the nAChR α2, α5, α6, α7, α10, and β2 subunits are expressed in MNLs, dendritic cells, and macrophages, while expression of mRNAs encoding the α4, α9, and β4 subunits varies [75,97]. These findings highlight the need for studies carried out under different physiological and/or pathophysiological conditions, as patterns of mRNA expression and levels of each nAChR subunit likely vary depending on the immunological status of the subject. Therefore, the lack of AChR mRNA expression in some of human immune cell specimens does not necessarily imply complete absence, but may be deeply suppressed, perhaps due to the plasticity of AChR mRNA expression resulting from immune response [103].

### 4.3. Role of α7 nAChRs in Regulation of Immune Function and Inflammatory Responses

As mentioned above, among the various subtypes of nAChRs expressed in immune cells, the contributions of homopentameric α7 nAChRs to inflammation and immune cell function are currently major topics of investigation [104]. Among the wide variety of nAChR subtypes, α7 nAChRs are expressed in most immune cells, including macrophages, dendritic cells, T cells and B cells [75,97]. In macrophages, for example, α7 nAChRs have been shown to negatively control synthesis and release of proinflammatory cytokines such as TNF-α, IL-1β and IL-6 [105]. This suggests cholinergic regulation of immune cell function may be mediated, at least in part, via α7 nAChR-driven pathways. Moreover, evidence suggests the potential utility of α7 nAChR agonists as immunomodulatory agents [104,106,107].

#### 4.3.1. Role of α7 nAChRs in the Regulation of Antibody Production

Upon immunization with OVA, α7 nAChR subunit gene-deficient (α7-KO) mice exhibit higher plasma concentrations of antigen-specific IgG_1_ antibody than WT mice [108]. In addition, splenocytes from α7-KO mice immunized with OVA produce greater amounts of the pro-inflammatory cytokines TNF-α, IFN-γ, and IL-6 than splenocytes from C57BL/6J WT mice immunized with OVA (Fujii et al., 2007) [108]. α7 nAChRs and other nAChR subunits are also reportedly involved in the regulation of B cell development and activation [109]. These findings suggest that α7 nAChRs are involved in regulating proinflammatory cytokine production, which in turn modulates antibody production.

#### 4.3.2. Role of α7 nAChRs in the Regulation of T Cell Differentiation and Cytokine Production

Using spleen cells containing macrophages and T cells from OVA-specific TCR transgenic DO11.10 mice, Mashimo et al. [107] investigated the effects of the selective α7 nAChR agonist GTS-21 on the differentiation of CD4^+^ T cells. Upon activation of spleen cells with OVA, GTS-21 suppressed the differentiation of CD4^+^ T cells into regulatory T cells (Tregs) and effector T cells (Th1, Th2 and Th17) and downregulated production of IL-2, IFN-γ, IL-4, IL-17, and IL-6. By contrast, upon activation of spleen cells in an antigen processing-independent manner using the antigen epitope OVA peptide_323–339_ (OVAp), GTS-21 promoted the differentiation of CD4^+^ T cells into Tregs and effector T cells and upregulated production of the aforementioned cytokines. GTS-21 also promoted differentiation of TCR-activated CD4^+^ T cells into Tregs and effector T cells in WT C57BL/6J mice, but had no effect on the differentiation of activated CD4^+^ T cells in α7-KO mice. α7 nAChRs thus appear to be involved in the promotion of CD4^+^ T cell differentiation. Taken together, the findings summarized in this section suggest (1) that α7 nAChRs expressed in innate and adaptive immune cells play distinct roles in immune regulation; (2) that α7 nAChRs on antigen presenting cells such as macrophages and dendritic cells suppress CD4^+^ T cell (adaptive immune cells) activation by interfering with antigen presentation through inhibition of antigen processing; and (3) that α7 nAChRs on CD4^+^ T cells upregulate their differentiation into Tregs and effector T cells. These divergent roles of α7 nAChRs on antigen presenting cells and T cells may contribute to the modulation of immune response intensity.

Stimulation of α7 nAChRs with nicotine suppresses differentiation of naïve CD4^+^ T cell activated with anti-CD3/CD28 Abs into Th1 and Th17 cells, but enhances differentiation into Th2 cells [110,111]. On the other hand, Galitovskiy et al. [112] showed that in oxazolone colitis, nicotine acts via α7 nAChR-mediated pathways to increase the percentage of colonic Tregs while reducing Th17 cells, and that nicotine increases numbers of Tregs among CD4^+^ CD62L^+^ T cells activated with anti-CD3/CD28 Abs. These findings suggest that α7 nAChR signaling is involved in regulating immune function through modification of T cell activities such as differentiation and cytokine production.

#### 4.3.3. Role of α7 nAChRs in the Promotion of Human CD4^+^ T Cell Differentiation into Tregs

Along with *CHRNA7*, the gene encoding the normal α7 nAChR subunit, a human-specific partially duplicated gene, *CHRFAM7A*, which lacks coding for the subunit’s ligand binding region, is also present on chromosome 15 [113]. Their expression yields two different mRNAs for α7 subunits, and their translation produces the normal α7 subunit as well as a mutated dup7 subunit lacking the ligand-binding site [114]. Consequently, in humans, α7 nAChRs composed of pentameric α7 and dupα7 subunits in varying proportions can be formed. α7 nAChRs with a large dupα7 composition may not function well as ion channels, as dupα7 acts as a dominant negative regulator of ion channel function [115,116,117,118,119,120]. These differences in the structure of human and animal α7 nAChRs complicate the generalization of α7 nAChR pharmacological data from animals to humans.

α7 nAChRs have been shown to have dual functions as canonical ionotropic channels and non-canonical metabolic signaling receptors in both neuronal and non-neuronal cholinergic cells [121]. In immune cells, α7 nAChRs appear to function as metabotropic rather than as ionotropic receptors [122,123,124]. α7 nAChRs with metabotropic function are coupled to heterotrimeric G proteins such as Gαq. Upon activation by a ligand, metabotropic α7 nAChR function induces release of G proteins, which then bind to the G protein-binding cluster in the M3-M4 loop of the channel and activate signaling cascades to mobilize Ca^2+^ from intracellular stores [125,126]. Because dupα7 retains the M3-M4 loop, once at least one intact α7 subunit contained in the α7 nAChR is activated by a ligand, dupα7 should, theoretically, contribute to the metabotropic receptor function. However, the functional effects of dupα7 subunits contained within metabotropic α7 nAChRs on immune cells are not yet known.

As described above, animal studies have shown that α7 nAChRs expressed in CD4^+^ T cells are involved in promoting differentiation into Tregs [106,107,112,127,128]. However, the finding that human peripheral blood leukocytes express more *CHRFAM7A* than *CHRNA7* [120,129,130] indicates the necessity for translational studies using human specimens to confirm the efficacy of compounds found to be effective in animal studies.

To address that issue, Mashimo et al. [131] investigated the mRNA expression of both the α7 and dupα7 subunits in human CD4^+^ T cells and the effect of the α7 nAChR agonist GTS-21 on Treg differentiation. Varying levels of α7 and dupα7 subunit mRNA were detected in human CD4^+^ T cells obtained from 15 subjects of different ethnic origins (Figure 3A), and no clear trends in the mRNA expression of the α7 and dupα7 subunits were observed across gender, age, or ethnicity. The cause of the large interindividual variation in α7 and dupα7 subunit expression is not yet clear, but genetic disposition, immunological regulation, or both may be responsible. Moreover, the greater interindividual variation in expression of the α7 than dupα7 subunit suggests that α7 nAChRs in CD4^+^ T cells are more susceptible to immune stimulation in daily life.

TCR-activation in T cells promotes ChAT mRNA expression, resulting in increased ACh synthesis and release [72]. Mashimo et al. [131] observed that following TCR activation in T cells, *CHRNA7* expression was profoundly suppressed on days 4 and 7 as compared to day 1 (Figure 3B). This suggests that *CHRNA7* expression may be suppressed by a negative feedback mechanism activated by the continuous α7 nAChR activation elicited by ACh released from T cells. By contrast, *CHRFAM7A* expression was unaffected by the negative feedback, and remained nearly constant. The different expression patterns of *CHRNA7* and *CHRFAM7A* can be explained by the finding that *CHRNA7* and *CHRFAM7A* are independently regulated by their respective promoters [120,129,132].

mRNA expression of the α7 and dupα7 nAChR subunits in TCR-activated T cells was not further affected by GTS-21 (Figure 3B(2)). However, GTS-21 did promote Treg development to varying degrees in samples from all individuals on days 5 (Figure 3C). These results suggest the potential ex vivo utility of GTS-21 for rapid preparation of large numbers of Tregs for adaptive immunotherapy, even with high expression of the dupα7 subunit. Moreover, the ex vivo utilization of GTS-21 should help reduce the time and associated costs of preparing sufficient numbers of Tregs.

### 4.4. Cholinergic Anti-Inflammatory Reflex

Efferent vagus nerve stimulation saves rats from lethal septic shock induced by intraperitoneal administration of the bacterial endotoxin lipopolysaccharide, inhibits hepatic synthesis of the proinflammatory cytokine TNF-α and suppresses increases in serum TNF-α concentrations [133]. In addition, Wang et al. [105] found that activation of α7 nAChRs in LPS-stimulated macrophages inhibits TNF-α synthesis, suggesting the involvement of ACh in an efferent anti-inflammatory reflex circuit. It was initially proposed that, within this reflex circuit, postganglionic efferent cholinergic nerves stimulated by preganglionic vagal efferent directly innervated macrophages, and ACh released from the nerve terminals acted on α7 nAChRs on macrophages, thereby suppressing TNF-α synthesis [134]. However, the splenic nerve, which innervates the spleen, is catecholaminergic [135], and no cholinergic innervation of the spleen has ever been detected [8]. As originally conceived, therefore, this neural circuit lacks a source of ACh to act on α7 nAChRs in macrophages in the spleen. The missing link was subsequently identified as a subset of ACh-producing splenic T cells that release ACh upon vagus nerve stimulation, leading to suppression of TNF-α production in macrophages [136]. This seems reasonable, as catecholamines have been shown to act on β adrenoceptors on T cells to increase ACh synthesis and release by promoting ChAT gene expression via cAMP/PKA pathways [79,80] (see Section 4.1.3). These findings and the accumulated evidence indicate that stimulation of α7 nAChRs by ACh derived from T cells induces activation of the JAK2/STAT3 signaling cascade in macrophages and inhibits NF-κB-mediated activation of TNF-α transcription, resulting in inhibition of TNF-α synthesis [137,138].

However, because norepinephrine (NE) is far more stable than ACh in the circulation, we cannot exclude the possibility that NE released from activated sympathetic nerve terminals not only acts on β_2_ adrenoceptors in T cells to promote ACh release, but also attenuates TNF-α synthesis in macrophages [135,139]. Determination of the precise mechanism underlying the anti-inflammatory reflex will require further investigation.

## 5. Conclusions

Studies of non-neuronal cholinergic systems recall the fact that the first discovery of ACh in an animal’s body was in the spleen, which lacks cholinergic innervation. There is now evidence not only that ACh is expressed in all life forms on earth, but also that non-neuronal cholinergic systems play key roles in widely varied aspects of the physiology and pathophysiology of animals, including humans. α7 nAChR expressed in macrophages and T cells is involved in the regulation of immune function and inflammatory responses, making it a potential therapeutic target.

## Figures and Tables

**Figure 1 ijms-25-04564-f001:**
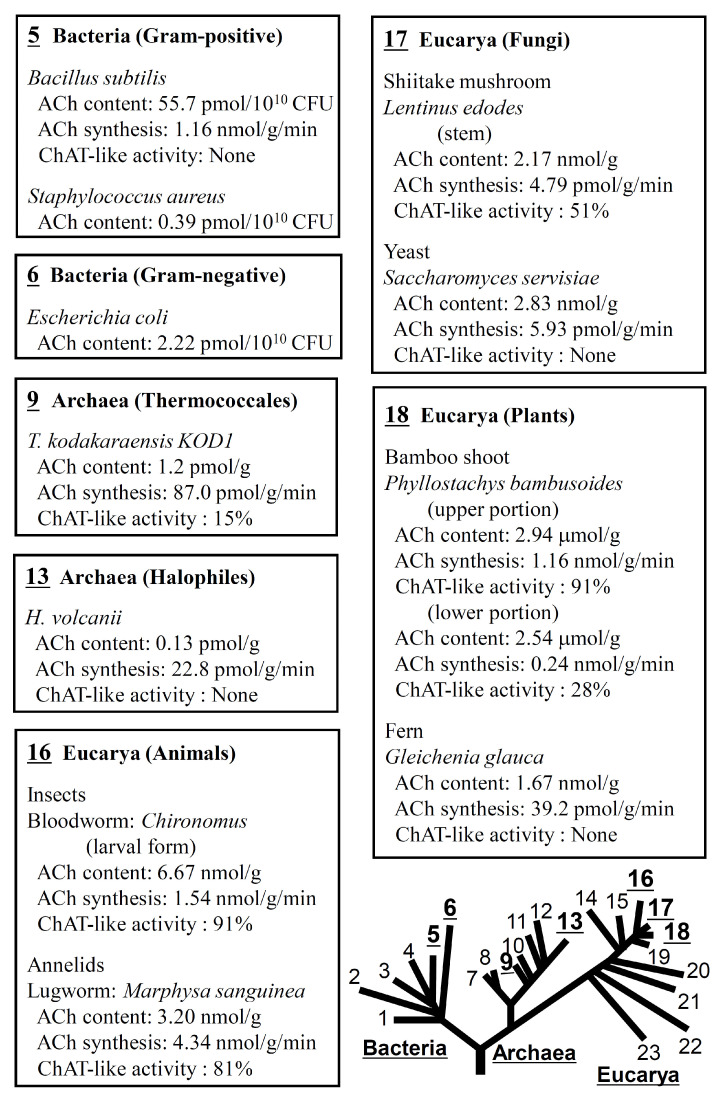
Expression of ACh and ACh-synthesizing activity in representative life forms and the rooted universal phylogenetic tree adopted from Wheelis et al. [33]. Bacteria: 1, Thermotogales; 2, flavobacteria and relatives; 3, cyanobacteria; 4, purple bacteria; 5, Gram-positive bacteria; and 6, green nonsulfur bacteria. Archaea-kingdom Crenarchaeota: 7, the genus Pyrodictium; and 8, the genus Thermoproteus; and Archaea-kingdom Eurycarhaeota: 9, Thermococcales; 10, Methanococcales; 11, Methanobacteriales; 12, Methanomicrobailes; and 13, extreme halophiles. Eurycarya: 14, entamoebae; 15, slim molds; 16, animals; 17, fungi; 18, plants; 19, ciliates; 20, flagellates; 21, trichomonads; 22, microsporidia; 23, diplomonads. ACh, acetylcholine; CFU, colony forming unit; ChAT, choline acetyltransferase. The ACh content was determined using an RIA for ACh [11]. ACh synthesis was determined using a modification of the procedure of Fonnum in the presence of 0.15 mM acetyl-CoA and 15 mM choline [14]. ChAT-like activity was calculated as the ratio of ACh synthesis sensitive to the selective ChAT inhibitor bromoacetylcholine (BrACh) to the total ACh synthesis. Arranged from data presented in references [31,32,34].

**Figure 2 ijms-25-04564-f002:**
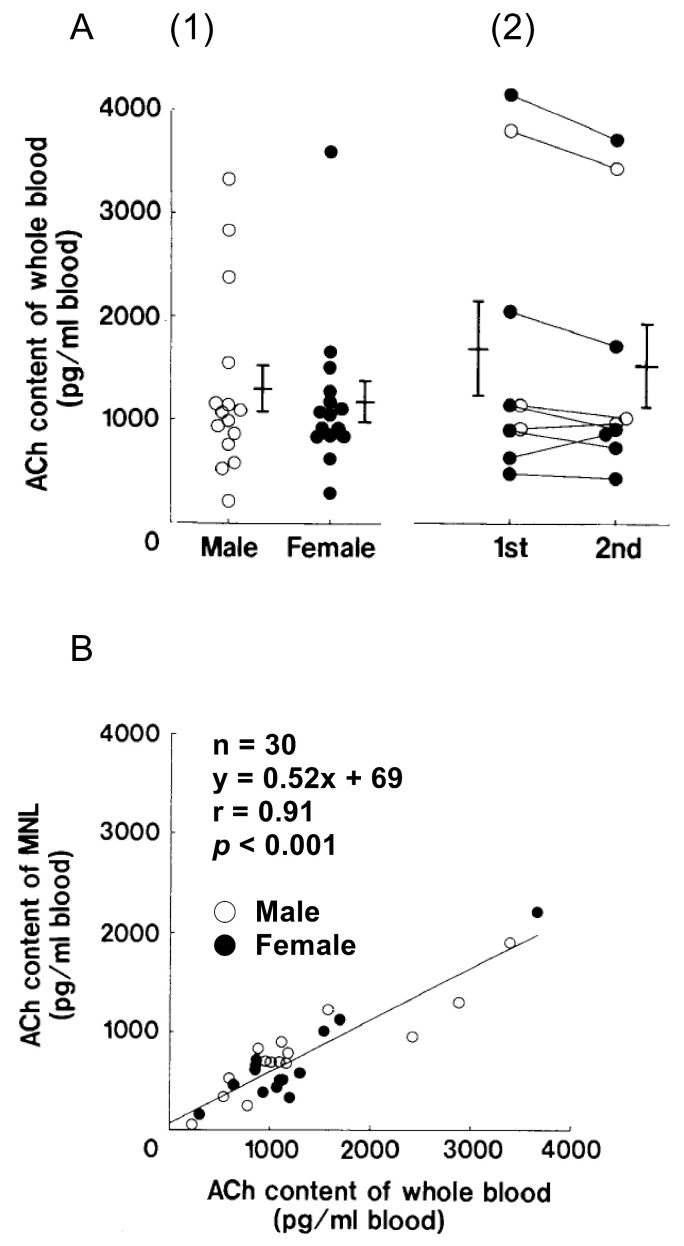
(**A**) (1) ACh contents of whole blood from 15 male (◦) and 15 female subjects (●). (2). Stability of blood ACh content in humans determined on two different occasions. The second measurement was performed 6 to 24 months after the first. Each bar represents the mean ± S.E.M. (**B**) Scatterplot showing the relation between the ACh content of whole blood and that of the MNLs. About 60% of the whole blood ACh content was localized in the MNL fraction. Arranged from data presented in a reference [70].

**Figure 3 ijms-25-04564-f003:**
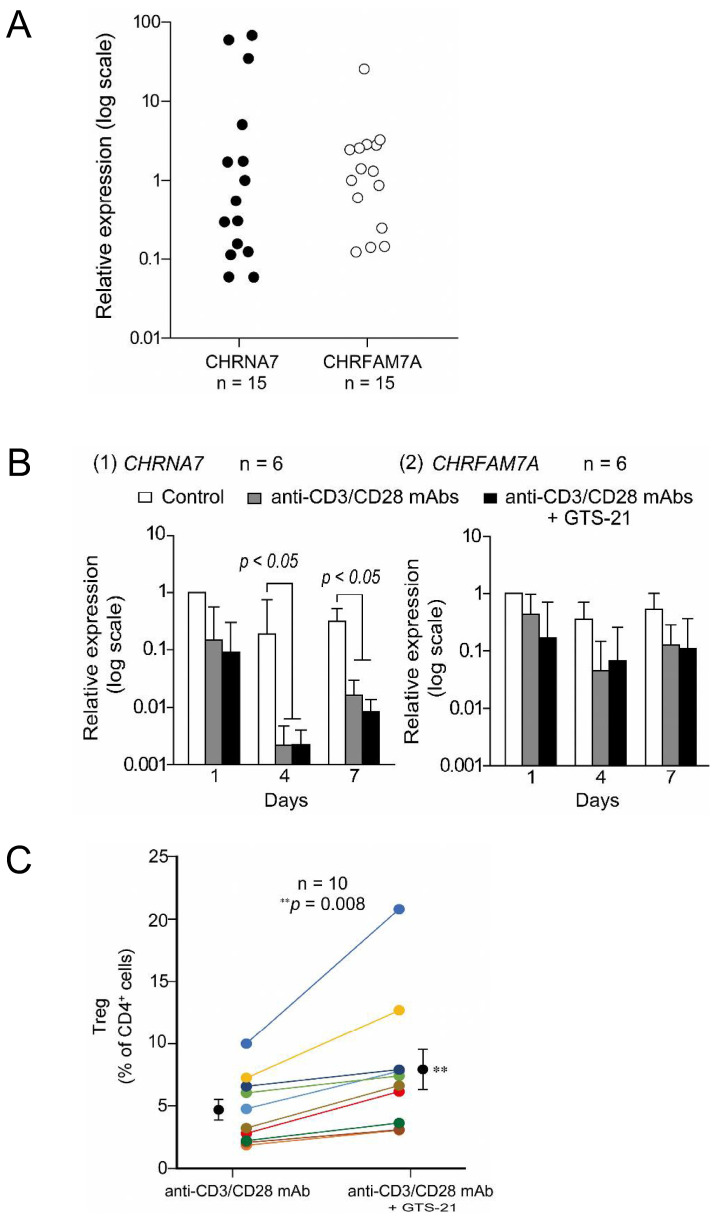
(**A**). Expression of α7 and dupα7 subunit mRNAs in resting human CD4^+^ T cells. Expression levels of *CHRNA7* and *CHRFAM7A* (α7 and dupα7 subunit mRNA, respectively) were first normalized to *GAPDH* mRNA in each individual. Then, to compare the magnitude of interindividual variability in *CHRNA7* and *CHRFAM7A* expression, levels of *CHRNA7* and *CHRFAM7A* mRNA were divided by the values closest to their respective medians and plotted on a logarithmic scale. The interindividual variability of *CHRNA7* expression was statistically greater than that of *CHRFAM7A* expression (F_(14, 14)_ = 2.18 × 10^−5^). (**B**). Fluctuations in the mRNA expression of the α7 and dupα7 subunits during TCR activation. Human CD4^+^ T cells were cultured for up to 7 days in the standard culture medium in the presence or absence of human T-activator CD3/CD28 Dynabeads with or without 30 μM GTS-21. Levels of *CHRNA7* and *CHRFAM7A* expression in cells from each individual was first normalized to *GAPDH* expression. Then, to detect fluctuations over time induced by TCR activation, *CHRNA7* (1) and *CHRFAM7A* (2) mRNA levels were further divided by their respective levels in controls observed on day 1. Bars are the mean ± S.E.M. (n = 6). Statistical significance was assessed with two-way ANOVA and post hoc Tukey tests. (**C**). Effects of GTS-21 on Treg development. GTS-21 enhanced Treg development from TCR-activated human CD4^+^ T cells on day 5 of culture. Gates were used to calculate the percentages of Tregs (CD4^+^CD25^+^FoxP3^+^ cells). For comparison, a line connects the percentage of Tregs observed in the absence and presence of 30 μM GTS-21 among cells from the same individuals. Bars are means ± S.E.M. (n = 10). Statistical significance was assessed using paired *t*-tests (** *p* < 0.01). Arranged from data presented in a reference (Mashimo et al., 2023) [131].

**Table 1 ijms-25-04564-t001:** Expression of non-neuronal ACh or ChAT in respective mammalian tissues and cells.

1. Cancer cells
(1) Lung cancer cells [42]
(2) Colon cancer cells [43]
(3) Stomach cancer cells [44]
2. Cardiovascular cells
(1) Cardiomyocytes [45]
(2) Vascular endothelial cells [46,47]
3. Immune cells (T cells, B cells, and Monocytes) [48,49]
4. Digestive epithelial cells
(1) Gingival and esophageal epithelial cells [50]
(2) Small intestinal epithelial cells [51]
5. Reproductive organs
(1) Amniotic membrane [52]
(2) Placenta [53]
6. Respiratory epithelial cells
(1) Bronchial epithelial cells [51]
7. Myogenic cells and tendon
(1) Myogenic cells [54]
(2) Tendon [55]
8. Skin
(1) Keratinocytes [41]; see also a review by Kurzen et al. [56]
9. Urinary bladder [57,58]

**Table 2 ijms-25-04564-t002:** Expression of mRNAs encoding mAChRs and nAChRs in human immune cells. (**A**) Expression of mRNAs Encoding mAChR Subtypes. (**B**) Expression of mRNAs Encoding nAChR Subunits.

**(A) mAChR**						
**Sample**	**Cell Type**	**M_1_**	**M_2_**	**M_3_**	**M_4_**	**M_5_**
1 (F)	MNLs	+	+	+	+	+
2 (F)	MNLs	−	+	−	+	+
3 (F)	MNLs	+	+	+	+	+
4 (F)	MNLs	+	−	+	+	+
5 (M)	MNLs	+	+	−	+	+
6 (M)	MNLs	+	−	+	+	+
7 (M)	MNLs	−	+	+	+	+
**(B) nAChR**						
**Sample**	**Cell Type**	**α3**	**α5**	**α7**	**α9**	**α10**
1 (F)	T	+	+	+	+	+
	B	+	+	+	−	+
2 (F)	T	+	+	−	+	−
	B	+	+	+	+	+
3 (F)	T	+	+	−	−	−
	B	+	+	+	−	−
4 (F)	T	+	+	+	−	−
	B	+	+	+	+	+
5 (F)	T	+	+	+	+	−
	B	+	−	−	+	+
6 (F)	T	−	+	−	+	−
	B	+	+	+	+	+
7 (F)	T	+	+	+	+	+
	B	−	+	+	+	+
8 (F)	T	−	+	+	+	+
	B	+	+	+	+	+

F, female; M, male. MNLs, mononuclear leukocytes. The mRNA expression for each AChR subunit or subtype was detected by amplifying cDNA samples for 40 cycles using reverse transcription polymerase chain reaction with specific order and reverse primers. +, positive expression; −,negative expression. Arranged from the data by Kawashima et al. [98].

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
