# Peer review of "Contributions of Non-Neuronal Cholinergic Systems to the Regulation of Immune Cell Function, Highlighting the Role of α7 Nicotinic Acetylcholine Receptors"

_ijms, 2024, doi:10.3390/ijms25084564_

Round 1
Reviewer 1 Report
Comments and Suggestions for Authors
In this review, the authors summarize the history of non-neuronal ACh and specifically focus on its immunomodulating actions, for example, on T cells and macrophages. This review could be comprehensive for readers who are not familiar with this field.
However, as a major comment, this review needs to add more information on why non-neuronal ACh exerts an anti-inflammatory effect through alpha7 receptors in terms of the molecular or DNA levels (section 4.4). As one of the mechanisms, they raise that alpha7 agonist enhances Treg differentiation in an antigen-processing-independent manner, unlike OVA alone, as cited from no. 111 reference. In this case, it will be more informative to add an explanation of why the difference occurs (section 4.3.2.)
As minor comments,
1. In line 52 in Page 2, "stored at -20 4.5 but C in a freezer"
The authors need to revise this part.
2. In Table 1 in Page 5, "7. Skeletal muscle [55]"
This reference no.55, as the title demonstrated, skeletal muscle may be not used in this study, but the synovial layers. They need to exchange a proper reference, instead.
Author Response
Responses to Reviewer #1 Comments-240411
First of all, we would like to thank the reviewer for these very insightful comments.
Major
Point 1: However, (1) as a major comment, this review needs to add more information on why non-neuronal ACh exerts an anti-inflammatory effect through alpha7 receptor in terms of the molecular or DNA levels (section 4.4). As one of the mechanisms, they raise that alpha7 agonist enhances Treg differentiation in an antigenprocessing-independent manner, unlike OVA alone, as cited from no. 111 reference. (2) In this case it will be more informative to add an explanation of why the difference occurs (section 4.3.2.).
Response 1-1: In accordance with the reviewer’s suggestion, we have inserted a sentence (shown below) in Section 4.4, lines 484-488, explaining at the molecular level why non-neuronal ACh exerts an anti-inflammatory effect through alpha7 receptors.
“These findings and the accumulated evidence indicate that stimulation of a7 nAChRs by ACh derived from T cells induces activation of the JAK2/STAT3 signaling cascade in macrophages and inhibits NF-κB-mediated activation of TNF-a transcription, resulting in inhibition of TNF-a synthesis [137,138].“
- de Jonge, W.J.; van der Zanden, E.P.; The, F.O.; Bijlsma, M.F.; van Westerloo, D.J.; Bennink, R.J.; Berthoud, H.R.; Uematsu, S.; Akira, S.; van den Wijngaard, R.M.; et al. Stimulation of the vagus nerve attenuates macrophage activation by activating the Jak2-STAT3 signaling pathway. Nat Immunol 2005, 6, 844-851, doi:10.1038/ni1229.
- Parrish, W.R.; Rosas-Ballina, M.; Gallowitsch-Puerta, M.; Ochani, M.; Ochani, K.; Yang, L.H.; Hudson, L.; Lin, X.; Patel, N.; Johnson, S.M.; et al. Modulation of TNF release by choline requires alpha7 subunit nicotinic acetylcholine receptor-mediated signaling. Mol Med 2008, 14, 567-574, doi:10.2119/2008-00079.Parrish.
Response 1-2: In accordance with the reviewer’s suggestion, we have modified a part of the description in lines 370-375 in Section 4.3.2. to explain the differences in the effects of alpha7 nAChR stimulation between innate and adoptive immune cells. Please see below.
“Taken together, the findings summarized in this section suggest 1) that α7 nAChRs expressed in innate and adoptove immune cells play distinct roles in immune regulation; 2) that α7 nAChRs on antigen presenting cells such as macrophages and dendritic cells suppress CD4+ T cell activation by interfering with antigen presentation through inhibition of antigen processing; and 3) that α7 nAChRs on CD4+ T cells (adomptive immune cells) upregulate their differentiation into Tregs and effector T cells.”
Minor
Point 1: In line 52 in Page 2, “stored at -20 4.5 but C in a freezer''. The authors need to revise this part.
Response 1: Thank you for pointing out the typo. We have corrected to read as “at -20°C in a deep freezer” in line 53.
Point 2: In Table 1 in Page 5. “7. Skeletal muscle [55]”. This reference no. 55, as the title demonstrated, skeletal muscle may be not used in this study, but the synovial layers. They need to exchange a proper reference, instead.
Response 2: We would like to thank the reviewer for pointing out the error in Table 1, #7, Skeletal muscle and tendon. #7 was replaced with “7. Myogenic cells and tendon, 1) Myogenic cells [54]” Accordingly, previous reference [47] has been replaced by [54]: Hamann et al (1995).
[54] Hamann, M.; Chamoin, M.C.; Portalier, P.; Bernheim, L.; Baroffio, A.; Widmer, H.; Bader, C.R.; Ternaux, J.P. Synthesis and release of an acetylcholine-like compound by human myoblasts and myotubes. J Physiol 1995, 489, 791–803, doi: 10.1113/jphysiol.1995.sp021092.
Reviewer 2 Report
Comments and Suggestions for Authors
This is a nice review by authors who have substantially contributed to the field.
Line 318: Please comment on the finding that mRNA for alpha10 subunits is sometimes found in the absence of alpha9 subunits. Is there evidence that alpha10 can form functional receptors in the absence of alpha9 ? Might alpha10 combine with other nAChR subunits?
Line 333: In addition to TNF-alpha, what other cytokines may be modulated by alpha7 nAChRs in macrophages?
368: What concentration of nicotine is needed to suppress differentiation of T-cells. Is this relevant to concentrations of nicotine experienced by those who smoke or vape?
486: The authors mention in the abstract and conclusion that the e involvement of α7 nAChRs 4expressed macrophages and T cells in regulating immune function and inflammatory responses make them potential therapeutic targets.
488: …’make them are’ please reword.
Author Response
Responses to Reviewer #2-240411
We would like to thank the reviewer for these very insightful comments.
Point 1: Line 318: Please comment on the finding that mRNA for alpha10 subunits is sometimes found in the absence of alpha9 subunits. Is there evidence that alpha10 can form functional receptors in the absence of alpha9 ? Might alpha10 combine with other nAChR subunits?
Response 1: First, we have inserted a following sentence in the caption of Table 1, which describes the procedure for detecting mRNA expression of each AChR subunit or subtype in the immune cell samples from normal volunteers.
“The mRNA expression for each AChR subunit or subtype was detected by amplifying cDNA samples for 40 cycles using reverse transcription polymerase chain reaction with specific order and reverse primers.” Therefore, if the mRNA expression for α9 is very weak, the table may show that α9 subunit is absent. We would like to point out that in human immune cell samples, large interindividual variations were observed not only in the α9 but also in the α7 subunit (Figure 3), probably due to interindividual differences in immunological status. Furthermore, the plasticity of mRNA expression for AChRs has been confirmed in TCR-activated T cells from mice (Qian et al, 2011 [103]). Therefore, we consider that the absence of AChR mRNA expression in some of human immune cell specimens does not necessarily mean that complete absence, rather may deeply suppressed probably due to immunological activation. Furthermore, as shown in Figure 2, interindividual variations in ACh content of blood and mononuclear leukocytes (MNLs) in general human population are also very large. To explain the plasticity of AChR mRNA expression, we have inserted a sentence in lines 330-333 to read as “Therefore, the lack of AChR mRNA expression in some of human immune cell specimens does not necessarily imply complete absence, but may be deeply suppressed, perhaps due to the plasticity of AChR expression resulting from immune response [103].”
Regarding α10 subunit, we do not have any information on the formation of α10 homopentameric nAChRs in T cells and B cells. However, based on what we know so far, α10 nAChRs are unlikely to be functional. Furthermore, there is currently no information regarding the formation of heteromeric nAChRs between α10 subunit and other nAChR subunits in T cells and B cells.
Point 2: Line 333: In addition to TNF-alpha, what other cytokines may be modulated by alpha7 nAChRs in macrophages?
Response 2: We would like to thank your comment suggesting us that reader would benefit from further elaboration on the cytokines regulated by alpha7 nAChRs in macrophage. So, we have rewritten the sentence in lines 340-342 to read as “In macrophages, for example, a7 nAChRs have been shown to negatively control synthesis and release of proinflammatory cytokines such as TNF-a, IL-1b and IL-6 [105]”.
Point 3: Line 368: What concentration of nicotine is needed to suppress differentiation of T-cells. Is this relevant to concentrations of nicotine experienced by those who smoke or vape?
Response 3: Information regarding the effect of nicotine on T cell differentiation was cited from references [108, 109] by Nizri et al (2009, 2012). According to Nizri et al [109], nicotine was administered sc to C57BL/6 mice at a dose of equivalent to 2 mg/kg/day for 28 days using ALZET miniosmotic pumps. Further information, such as plasma nicotine concentrations and effects of smoking on T cell differentiation, is not available from that reference. However, they raised the possibility that systemically administered nicotine induces immunomodulatory effects via elevation of corticosteroid release. Number of Citations: [108], 315; [109], 66, as of April 6, 2024.
Point 4: Line 486: The authors mention in the abstract and conclusion that the involvement of α7 nAChRs expressed macrophages and T cells in regulating immune function and inflammatory responses make them potential therapeutic targets.
“How you perceive therapies can be developed to target cholinergic pathways and do we already have anything available which could be used or repurposed for this purpose.?”
Response 4: Specific therapeutics targeting α7 nAChRs in T cells have not yet been developed. However, the results shown in Figure 3 demonstrate that, despite high expression of dupα7 in human T cells, the ex vivo utilization of the α7 nAChR agonist GTS-21 should help reduce the time and associated costs of preparing sufficient numbers of Tregs for adaptive immune therapy. α7 nAChRs in macrophages described in the Section 3.3. Cholinergic anti-inflammatory reflex should also be a potential target for therapeutic drug development.
Accumulating evidence indicates that not only nAChRs but also mAChRs are targets for therapeutic drug development in dermatology and oncology (Please see reviews by Grando et al [23, 24]).
Point 5: Line 488: …’make them are’ please reword.
Response to 5: Thank you for pointing out a typo. We have restated the sentence in lines 500-502, to read as “α7 nAChR expressed in macrophages and T cells is involved in regulation of immune function and inflammatory responses, making it a potential therapeutic target.”